# COVID-19 cross-sectional study in Maricá, Brazil: The impact of vaccination coverage on viral incidence

Thiago Silva Frauches[1][☉], Carlos Alberto de Senna Costa[2][☉], Claudia dos Santos Rodrigues[3], Marcelo Costa Velho Mendes de Azevedo[4,5], Michelle de Moraes Ferreira[5], Hanna Beatriz Vieira da Silva Ramos[2], Wilson Rodrigues de Souza Junior[1], Andréa Ribeiro Costa[1], Adriana Cardoso Camargo[1], Adriana Halfeld Alonso[1], Fábio Álvaro dos Santos[1], Hércules da Silva Oliveira[1], Janaína Guimarães Coelho[1], Joyce Florentina da Silva Sobral[1], Luciane Cardoso dos Santos Rodrigues[1], Marcio Martins Casaes Ferreira[1], Patricia Laureano[1], Raquel Adalgiza da Paz Fernandes[1], Renata da Silva Santos[1], Rose Mary Carvalho dos Santos[1], Sanderson Milagres[1], Vanessa Cristina Conceição dos Santos[1], Jussara Teixeira Silva[5], Tatiana Martins da Silva[5], Malu Gabriela Costa da Rocha[5], Andreia Edwirges de São Carlos[5], Amorim Mourão de Araújo Ramos[5], Fernanda Martins de Almeida Bastos[5], Daina Raylle Francisco[5], Sabrina dos Santos Rosa[5], Layla Corrêa Linhares[5], Raissa Rodrigues Organista[5], Leandro Bastos[5], Maria Magdalena Kelly Pinto[5], Jean Pablo Lima do Nascimento[6], João Pedro Moura da Silveira[6], Mateus Quintanilha dos Santos[6], Nathaly Santos da Silva[6], Nayra Cristina dos Santos Ferreira[6], Rafael Brito Ramirez Reis[6], Ruan Fonseca de Oliveira[6], Valdinei de Oliveira Sá[6], Thyago Ramos de Siqueira Hammes[6], Juliano de Oliveira Monteiro[6], Pedro Henrique Cardoso[7], Mônica Barcellos Arruda[7], Patricia Alvarez[7], Richard Araujo Maia[8], Liane de Jesus Ribeiro[8], Orlando Costa Ferreira, Jr[8], Aline Santos[9], Alberto Carlos Melo de Almeida[9], Lauro Garcia[10], Celso Pansera[1], Amilcar Tanuri[2,8]*

1 Laboratório Central Dr. Rímolo Neto/LACEN, Secretaria Municipal de Saúde, Prefeitura Municipal de Maricá, Maricá, Rio de Janeiro, Brazil, 2 Instituto de Ciência, Tecnologia e Inovação de Maricá/ICTIM, Maricá, Rio de Janeiro, Brazil, 3 Centro de Testagem e Aconselhamento, Secretaria Municipal de Saúde, Prefeitura Municipal de Maricá, Maricá, Rio de Janeiro, Brazil, 4 Universidade do Rio de Janeiro—UNIRIO, Rio de Janeiro, Brazil, 5 Secretaria Municipal de Saúde, Prefeitura Municipal de Maricá, Maricá, Rio de Janeiro, Brazil, 6 Secretaria de Cultura e Direitos Humanos, Prefeitura Municipal de Maricá, Maricá, Rio de Janeiro, Brazil, 7 Instituto de Tecnologia de Imunobiológicos Bio-Manguinhos, Fundação Oswaldo Cruz/Fiocruz, Rio de Janeiro, Brazil, 8 Laboratório de Virologia Molecular, Instituto de Biologia, Departamento de Genética, Universidade Federal do Rio de Janeiro/UFRJ, Rio de Janeiro, Brazil, 9 Brasilis, São Paulo, Brazil, 10 Laboratório Blessing, Maricá, Rio de Janeiro, Brazil

☉ These authors contributed equally to this work.
* atanuri1@gmail.com

**Data Availability Statement:** All relevant data are within the paper and its Supporting Information files.

## Abstract

Population surveillance in COVID-19 Pandemic is crucial to follow up the pace of disease and its related immunological status. Here we present a cross-sectional study done in Maricá, a seaside town close to the city of Rio de Janeiro, Brazil. Three rounds of study sampling, enrolling a total of 1134 subjects, were performed during May to August 2021. Here we show that the number of individuals carrying detectable IgG antibodies and the neutralizing antibody (NAb) levels were greater in vaccinated groups compared to unvaccinated ones, highlighting the importance of vaccination to attain noticeable levels of populational immunity against SARS-CoV-2. Moreover, we found a decreased incidence of COVID-19 throughout the study, clearly correlated with the level of vaccinated individuals as well as the

**Funding:** The funders had no role in study design, data collection and analysis, decision to publish, or preparation of the manuscript.

**Competing interests:** The authors have declared that no competing interests exist.

proportion of individuals with detectable levels of IgG anti-SARS-CoV-2 and NAb. The observed drop occurred even during the introduction of the Delta variant in Maricá, what suggests that the vaccination slowed down the widespread transmission of this variant. Overall, our data clearly support the use of vaccines to drop the incidence associated to SARS-CoV-2.

## Introduction

The coronavirus disease 2019 (COVID-19) pandemic reached the Latin America later than other continents [1, 2]. The first case recorded in Brazil dates back to February 25[th], 2020 [3]. In October 2021, Brazil accounted for the most cases and deaths in Latin America (>21 million cases and >600.000 deaths) [4]. Rio de Janeiro State concentrates 1.31 million cases and 67,000 deaths by the beginning of 41[st] epidemiological week [5]. Case incidence experienced a substantial decrease after large scale vaccination campaigns [5–7]. In fact, COVID-19 vaccination campaign in Rio de Janeiro State reached 80% of target population with at least one dose and 60% of fully vaccinated individuals by October 14[th], 2021 [5]. Until June 2021, Rio de Janeiro has experienced the circulation of three major variants in different time frames [8]. By the beginning of October 2020 there was the introduction of P2 (Zeta) variant of investigation (VOI), that was replaced by the beginning of 2021 by P1 (Gamma) variant of concern (VOC), which prevailed until June 2021 when Delta VOC arrived and dominated until beginning of 2022 [8].

The introduction of COVID vaccines in early 2021 has impacted the incidence of COVID-19 as well as the hospitalization and death associated with SARS-CoV-2 infections in different cohort studies [9–11]. Concurrently, The National Plan of COVID-19 Immunization in Brazil employed four vaccines on its strategy [6, 12]. The Brazilian campaigns first begun with the utilization of CoronaVac in January 2021, followed by AstraZeneca in February 2021 [6, 12]. On April 2021 Pfizer was included and for the last, Janssen was incorporated to the campaign strategies in June 2021 [6, 12].

Population-based data on COVID-19 are essential for guiding policies and evaluating public health interventions made in different cities [13–16]. However, there are few such studies, particularly from low or middle-income countries [15, 17]. Then, our aim is to investigate SARS-CoV-2 antibody (anti-SCOV2) prevalence and RT-PCR status in Maricá, a seaside town close to the city of Rio de Janeiro, Brazil. Maricá is located 60Km from the city of Rio de Janeiro and has a total population of 161,000 habitants. Since the beginning of COVID-19 pandemic, Maricá accounted for 18,657 cases and 584 deaths (mortality rate of 2.782/100,000 inhabitants) [7]. In this study, we disclose the results of three repeated cross-sectional COVID-19 seroprevalence and incidence surveillances from May to August 2021. For each round, samples from 384 individuals were randomly selected. Nasopharyngeal swabs and blood sera were collected to run RT-PCR targeting SARS-CoV-2 N gene and COVID-19 serology measurements such as neutralizing antibodies titles, respectively.

## Material and methods

### Sampling strategy

From May, 24[th] to August, 5[th] a multi-stage probabilistic sampling was adopted, with 39 census tracts selected with probability proportionate to size in each sentinel cross-sectional study,

and ten households at random in each tract. In order to select each census tracts maps and household listings made available by the Brazilian Institute of Geography and Statistics was utilized [18]. One individual was randomly selected from a listing of all household members. Subjects below 18 years old and those with mental disability or special needs were excluded. If the randomly selected person refused to provide sample or could not be found, the interviewers moved on to the next household on the right.

A questionnaire was applied to capture socio demographic and clinical data from all enrolled individuals. In addition, nasopharyngeal swab samples and 10mL of whole blood were collected by venipuncture to perform RT-PCR (swabs) and ELISA and serum neutralization antibodies titration (blood serum). Interviewers were equiped with all personel protective equipment required (aprons, gloves, surgical face masks, shoes and hair covers), discarded as hospital waste after each interview.

## Data and specimen collection

A smartphone application for data collection was used by interviewers for listing and selecting household members, and also to record answers. Participants answered short questionnaires on sociodemographic information (sex, age, education, and occupation) and compliance with physical distancing measures. Participants (and family) previous exposition to COVID-19 was also evaluated in the questionnaire. All selected participants were asked to sign an informed consent and a blood specimen was drawn for serological tests to estimate patients´ immunological status as well as a nasopharyngeal swab for RT-PCR COVID-19 molecular test to estimate the incidence of COVID-19 in each sampling cycle. See study raw data in S1 Data.

## Serological SARS-CoV-2 ELISA tests

To measure anti-SCOV2 RBD antibody levels, a chemiluminescent based immunoassay (CLIA) was performed with ACCESS SARS-CoV-2 IgM QC and ACCESS SARS-CoV-2 IgG II QC kits (Beckman Coulter, USA) in accordance with manufacturer instructions. Results were generated based on the ratio between the luminescence of tested specimen and the negative control. All results above 1.0 were considered positive in this assay.

To evaluate the title of neutralizing antibody in each sera specimens, Lumit SARS-CoV-2 Spike RBD:ACE2 immunoassay (Promega, Madison, WI, USA) was performed. Previously published protocol was followed and the result was calculated by the percentage of inhibition of RBD:ACE2 interaction by each serum analyzed. Inhibition above 70% was considered positive in terms the presence of neutralizing antibodies [19].

## Viral RNA extraction and RT-PCR test

Nasopharyngeal swab samples were pooled together–four samples per pooling [20]. Nucleic acid extraction was performed a in automated Maxwell® RSC platform (Promega, USA). Extract pools was shortly storage at 4˚C before RT-PCR analysis.

SARS-CoV-2 RNA detection was made following the CDC protocol for SARS-CoV-2 RT-PCR diagnosis (2019-nCoV CDC kit) [21] with CFX96 BioRad instrument. Pooled samples with detected Ct values in N1 and/or N2 were segregated and reanalyzed separately. Segregated nasopharyngeal swab samples were considered positive when Ct values for N1 and N2 were ≤ 38.

VOC assessments were made on SARS-CoV-2 RT-PCR positive samples by a 4Plex SARS-CoV-2 for VOC screening kit (Bio-Manguinhos, Brazil). The assay was based in a fourplex format. TaqMan probes for SARS-CoV-2 virus were used for detection a target region in the N gene, and screening samples with suggestive profiles for the different VOCs. Suggestive VOC

profiles were given by combining results obtained of the deletions (Del) S106, G107 and F108, in the ORF1a gene (NSP6) and Del. H69 and V70 in the Spike gene from the samples tested. Samples were considered positive when Ct values for SC2-N, Wt Del NSP6 and Wt Del 69, 70 were lower than 40.

## Ethics committee approval

Ethics approval was obtained from the UNIRIO Ethics Committee (CAAE 38341120.0.0000.5258), with written informed consent from all participants. Positive cases were reported to the municipal COVID-19 surveillance systems after participants agreed to the disclosure in the consent form.

## Data analysis

All data included in the patient´s questionnaire was saved in a database to perform the analysis. Sociodemographic data and its association with SARS-CoV-2 infections was done with Chi-square tests with Yates correction. Serological and NAb production groups correlations were done with a Mann-Whitney unpaired test. All graphics and statistical analysis were based on GraphPad Prism 9.0.0. software (GraphPad Software, LLC). P-values lower than 0.05 were considered significant. Vaccination effectiveness was calculated based on the ration between the incidence of SARS-CoV-2 infection (by RT-PCR status), in vaccinated compared to unvaccinated subjects.

## Results

During the three rounds of this study, a total of 1,134 subjects were interviewed. Table 1 resume overall collected sociodemographic data. Female participants were the majority during all three rounds (n = 679; 60%) as well as participants with age below 60 years old (yo) (n = 430; 38%). Thirteen percent of all participants showed previous COVID-19 diagnosis and

**Table 1. Overview of sociodemographic and epidemiological data in all three studies.**

| Characteristics | Participants | | | | | | | |
|---|---|---|---|---|---|---|---|---|
| | Round 1 | | Round 2 | | Round 3 | | Overall | |
| | (n = 363) | | (n = 384) | | (n = 387) | | (n = 1134) | |
| | % (no.) | Median (Range) | % (no.) | Median (Range) | % (no.) | Median (Range) | % (no.) | Median (Range) |
| **Gender** | | | | | | | | |
| Female | 59 (215) | | 63 (244) | | 57 (220) | | 60 (679) | |
| Male | 41 (148) | | 37 (140) | | 43 (167) | | 40 (455) | |
| **Age groups** | | | | | | | | |
| All participants | - | 54 (19–91) | - | 56 (18–91) | - | 54 (18–87) | - | 54 (18–91) |
| < 60 years old | 63 (230) | 42 (19–59) | 60 (230) | 46 (18–59) | 63 (244) | 43 (18–59) | 62 (704) | 43 (18–59) |
| ≥ 60 years old | 37 (133) | 68 (60–91) | 40 (154) | 66 (60–91) | 37 (143) | 68 (60–87) | 38 (430) | 68 (60–91) |
| **Previous COVID-19 diagnosis reported** | | | | | | | | |
| Participant | 13 (47) | | 13 (49) | | 13 (50) | | 13 (146) | |
| Family[a] | 31 (112) | | 29 (112) | | 18 (68) | | 26 (292) | |
| **Comorbidities reported** | | | | | | | | |
| Hypertension | 41 (149) | | 41 (157) | | 35 (136) | | 39 (442) | |
| Diabetes | 14 (51) | | 18 (68) | | 11 (41) | | 14 (160) | |
| Asthma/Bronchitis | 8 (28) | | 6,5 (25) | | (19) | | 6 (72) | |

[a]COVID-19 cases reported in relatives living in the same house.

almost one fourth of the interviewed participants reported disease in cohabiting relatives. This number increased to 40% when RT-PCR positive individuals were segregated; this correlation was statistically significant ($\chi2 = 5.1$; p = 0.02354).

The most prevalent comorbidity were hypertension, followed by diabetes, and then respiratory syndromes. Other sociodemographical characteristics such as educational level, hygiene and social distance compliance are detailed in S1 Table.

When all data regarding non-pharmacological measurements was analyzed, no differences between RT-PCR or anti-SCOV2 positive individuals and SARS-CoV-2 unexposed subjects (RT-PCR negative and anti-SCOV2 antibodies seronegative) were observed (S1 Table). Regarding the educational level, we found that RT-PCR positive results were higher in lower educational background. Moreover, we did not find positive cases in participants with superior educational levels (S1 Table).

Among RT-PCR positive individuals, main symptoms were cough and "body ache" (S1 Table). For seropositive individuals for COVID-19, main symptoms could not be distinguished from the general population and are related as running nose, cough, and headache.

The overall rate of positive SARS-CoV-2 RT-PCR (RT-PCR+) results was 1.76% (Fig 1A). We observed a progressive reduction of 58% in RT-PCR+ cases from the first to the third round of the study. The global Ct median of SARS-CoV-2 N1 target was 27.32 (range 16.61–37.04) and became stable across all three study rounds (Fig 1B). Our VOC screening analysis showed that in the first round 100% (n = 6) of RT-PCR+ of the samples had the deletion on H69 and V70 on Spike gene, an indicative of Gamma VOC profile. In the second round, six out of seven samples (85%,) had the same Gamma profile, with the remaining one presenting no deletions on ORF1a and Spike genes and being classified as "others". In the last round of the study, from three samples analyzed we found one with Gamma profile, one classified as "others" and one that showed deletions on (Del) S106, G107 and F108, in the ORF1a gene, and H69 and V70 on Spike gene, suggesting a Delta VOC SNP signature.

RT-PCR+ participants were predominantly female and below 60yo (Table 2). Seventy percent had comorbidities; hypertension was present in half of the participants followed by diabetes and respiratory syndromes. Regarding vaccination status, fifty five percent of RT-PCR + participants received at least one vaccine jab and 40% were fully immunized. Approximately 75% (n = 8) of vaccinated RT-PCR+ participants received at least one jab of CoronaVac. The remaining three individuals infected were immunized with AstraZeneca. Most of RT-PCR + participants reported recent symptoms related to COVID-19 (n = 13; 65%) and the remaining (n = 7; 33%) did not report any kind of symptoms. Fifty four percent (n = 7) of the symptomatic RT-PCR+ participants were vaccinated. Among them, five participants (71%) were fully immunized with CoronaVac and the remaining received two doses of AstraZeneca vaccine. No significant difference on N1 target Ct values (P = 0.94) was observed between infected vaccinated (M = 27.6) and unvaccinated (M = 27.0) individuals.

We observed an increase of 76% ($\chi2 = 98.9$; p<0.00001) in vaccinated participants across all three study rounds (Table 3). At the end of the third round, the global vaccination rate was 65% for participants receiving at least one vaccine jab and there no significant changes were found for fully immunized subjects. Over the three rounds, CoronaVac and AstraZeneca standard the most frequent vaccines administered.

The increase of vaccination rate impacted the anti-SCOV2 IgG (anti-SCOV2 IgG) serum levels on participants evaluated by CLIA assay. We observed a sustained increase of anti-SCOV2 IgG positive results through the three rounds (37%, 47% and 52%) with a overall rate of anti-SCOV2 IgG positive individuals of 48%. Fig 2 shows the anti-SCOV2 IgG profile in vaccinated and unvaccinated groups. Of note, both groups presented a rise in the number of anti-SCOV2 IgG positive individuals (Fig 2A). However, the percentage of anti-SCOV2 IgG

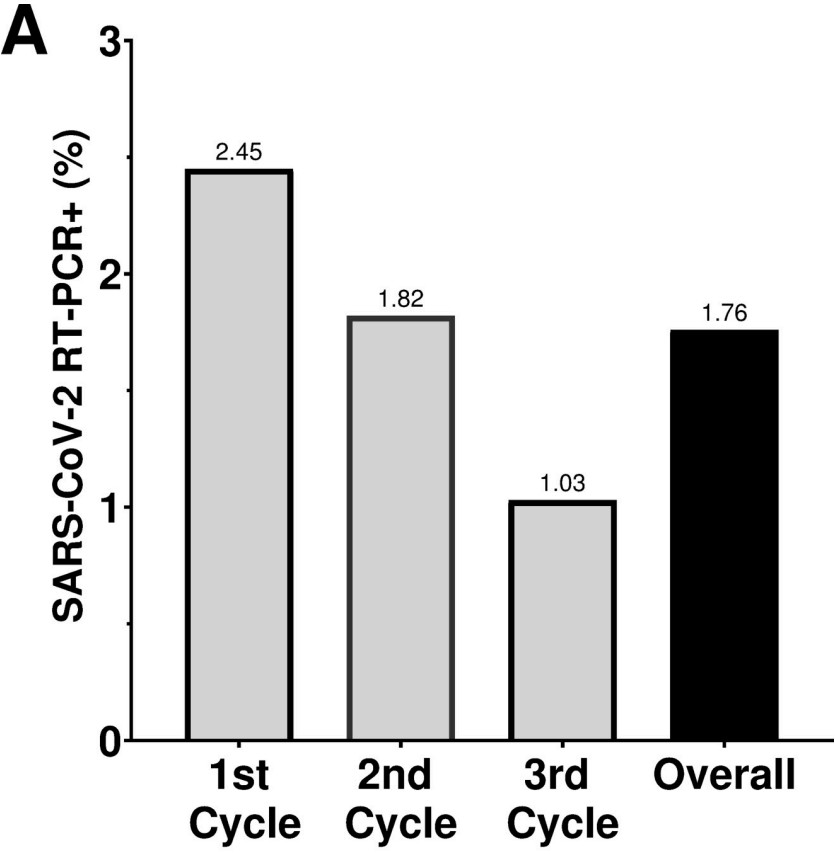

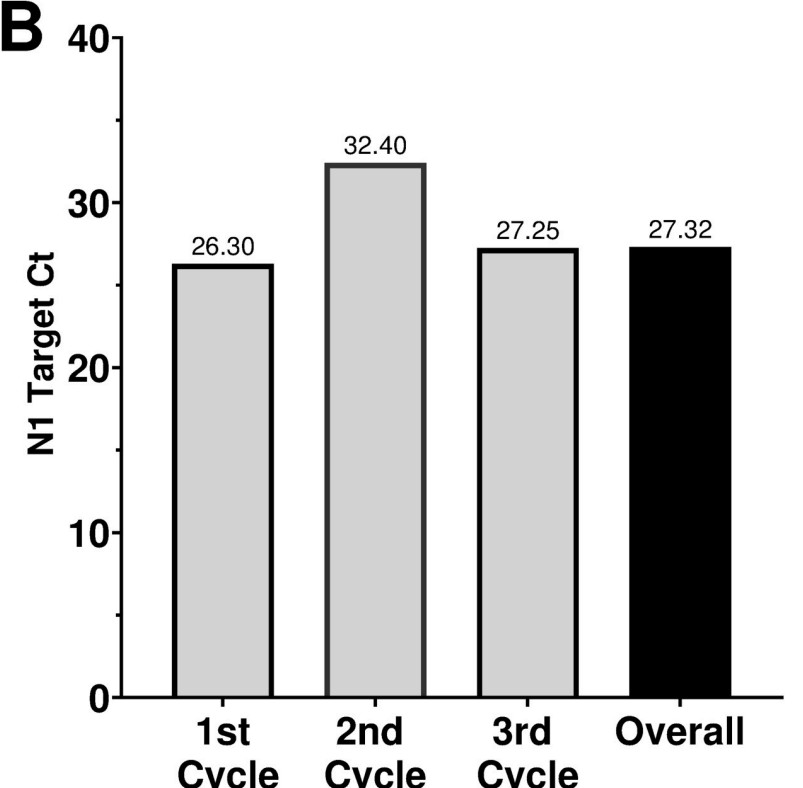

**Fig 1. Incidence of Covid-19 in the study.** A) Percentage of SARS-CoV-2 RT-PCR positive participants in all and each round of the study. B) N1 target Ct average of participants SARS-Cov-2 RT-PCR positive in all and each round of the study.

positive individuals was around three times higher in the vaccinated group on average. On the third round, the anti-SCOV2 IgG positive rate decreased in vaccinated individuals with age above 60yo (Fig 2C). Furthermore, anti-SCOV2 IgG positive rate in the third round increased to 47% on unvaccinated subjects (Fig 2A) and to 53% among unvaccinated subjects below 60yo (Fig 2B) when compared to the second round. This increase in the amount of unvaccinated IgG positive individuals in the $3^{rd}$ round was statistically significant ($\chi2 = 7.68$; $p = 0.05584$). When unvaccinated subjects carrying anti-SCOV IgG antibodies in the third round were analyzed, 78% of them reported no COVID-19 symptoms in the last 30 days prior to

**Table 2. Overview of all epidemiological and clinical data from RT-PCR positives individuals.**

| Characteristics | Participants | | | | | | | |
|---|---|---|---|---|---|---|---|---|
| | Round 1 | | Round 2 | | Round 3 | | Overall | |
| | % (no.) | Median (Range) | % (no.) | Median (Range) | % (no.) | Median (Range) | % (no.) | Median (Range) |
| | 100 (n = 9) | | 100 (n = 7) | | 100 (n = 4) | | 100 (n = 20) | |
| **Gender** | | | | | | | | |
| Female | 56 (5) | | 57 (4) | | 75 (3) | | 60 (12) | |
| Male | 44 (4) | | 43 (3) | | 25 (1) | | 40 (8) | |
| **Age groups** | | | | | | | | |
| All participants | - | 43 (27–70) | - | 55 (36–71) | - | 64 (34–76) | - | 50 (27–76) |
| < 60 years old | 67 (6) | 35 (27–45) | 57 (4) | 40 (36–55) | 25 (1) | 34 (34) | 55 (11) | 37 (27–55) |
| ≥ 60 years old | 33 (3) | 68 (67–70) | 43 (3) | 65 (63–71) | 7 5 (3) | 68 (60–76) | 45 (9) | 68 (60–76) |
| **Symptoms related to COVID-19** | | | | | | | | |
| Symptomatic | 67 (6) | | 71 (5) | | 50 (2) | | 65 (13) | |
| Asymptomatic | 33 (3) | | 29 (2) | | 50 (2) | | 35 (7) | |
| **Previous COVID-19 diagnosis** | | | | | | | | |
| Participant | 11 (1) | | 29 (2) | | 25 (1) | | 20 (4) | |
| Family[a] | 33 (3) | | 71 (5) | | 25 (1) | | 45 (9) | |
| **Comorbidities** | | | | | | | | |
| Hypertension | 44 (4) | | 43 (3) | | 75 (3) | | 50 (10) | |
| Diabetes | 22 (2) | | 0 (0) | | 0 (0) | | 10 (2) | |
| Asthma/Bronchitis | 22 (2) | | 0 (0) | | 0 (0) | | 10 (2) | |
| **Immunization status** | | | | | | | | |
| Unvaccinated | 67 (6) | - | 43 (3) | - | 0 (0) | - | 45 (9) | - |
| Partially immunized[b] | 0 (0) | 0 (0) | 29 (2) | 34 (27–40) | 25 (1) | 3 (3) | 15 (3) | 27 (3–40) |
| Fully immunized[c] | 33 (3) | 22 (21–44)[d] | 29 (2) | 58 (41–74) | 75 (3) | 79 (21–86) | 40 (8) | 43 (21–86 |
| | 100 (n = 3) | | 100 (n = 4) | | 100 (n = 4) | | 100 (n = 11) | |
| **Vaccine type** | | | | | | | | |
| AstraZeneca | 0 (0) | | 50 (2) | | 25 (1) | | 27 (3) | |
| CoronaVac | 100 (3) | | 50 (2) | | 75 (3) | | 73 (8) | |

[a]COVID-19 cases reported in relatives living in the same house.

[b]Individuals that received at least one vaccine dose.

[c]Individuals immunized with all doses preconized in the vaccine instruction insert.

[d]Days after last jab.

**Table 3. Overview of vaccination profile of all participants in the three rounds of the study.**

| Characteristics | Participants | | | |
|---|---|---|---|---|
| | Round 1 | Round 2 | Round 3 | Overall |
| | % (no.) | % (no.) | % (no.) | % (no.) |
| | **100 (n = 363)** | **100 (n = 384)** | **100 (n = 387)** | **100 (n = 1134)** |
| **Vaccination status** | | | | |
| Unvaccinated | 54 (196) | 32 (124) | 19 (74) | 35 (394) |
| Vaccinated[a] | 46 (167) | 68 (260) | 81 (313) [d] | 65 (740) |
| | 100 (n = 167) | 100 (n = 260) | 100 (n = 313) | 100 (n = 740) |
| **Immunization status** | | | | |
| Partially immunized[a] | 50 (84) | 63 (164) | 53 (165) | 56 (413) |
| Fully immunized[b] | 50 (83) | 34 (96) | 47 (148) | 44 (327) |
| **Vaccine type** | | | | |
| CoronaVac | 55 (92) | 38 (100) | 36 (114) | 41 (306) |
| AstraZeneca | 42 (71) | 53 (139) | 48 (149) | 49 (359) |
| Pfizer | 2 (3) | 8 (21) | 14 (44) | 9 (68) |
| Janssen | 0 (0) | 0 (0) | 2 (6) | 1 (6) |
| Mixed[c] | 1 (1) | 0 (0) | 0 (0) | <1 (1) |
| | 100 (n = 83) | 100 (n = 96) | 100 (n = 148) | 100 (n = 327) |
| **Fully immunization by vaccine type** | | | | |
| CoronaVac | 90 (75) | 94 (90) | 62 (91) | 78 (256) |
| AstraZeneca | 9 (7) | 6 (6) | 34 (51) | 20 (64) |
| Pfizer | 0 (0) | 0 (0) | 0 (0) | 0 (0) |
| Janssen | 0 (0) | 0 (0) | 4 (6) | 2 (6) |
| Mixed | 1 (1) | 0 (0) | 0 (0) | <1 (1) |

[a]Individuals that received at least one vaccine dose.

[b]Individuals immunized with all doses preconized in the vaccine instruction insert.

[c]First dose CoronaVac and second dose AstraZeneca.

[d] p<0.00001

interview, suggesting asymptomatic infections in this group. This number contrasts with the RT-PCR+ counterpart where most of the infections were symptomatic.

In general, the median title of anti-SCOV2 IgG in fully immunized individuals was higher than in unvaccinated individuals (Fig 3). In our study, 90% of vaccinated individuals received CoronaVac or AstraZeneca vaccines. Both vaccines produced significant levels of anti-SCOV2 IgG (p<0.0001) in fully immunized individuals when compared to unvaccinated ones, independent of age (Fig 3A–3C). Of note, there was no significant difference between one dose CoronaVac population (IgG level OD/Cut-off M = 0.2) and unvaccinated individuals. The median levels of anti-SCOV2 IgG in unvaccinated subjects was drastically lower (M = 0,08) when compared to fully vaccinated ones (CoronaVac: M = 1,17 and AstraZeneca: M = 4,19). Even when analyzed by age, CoronaVac (<60yo M = 0.97; ≥60yo M = 1.19) and AstraZeneca (<60yo M = 3.27; ≥60yo M = 4.62) fully immunized groups exhibited higher median levels compared to unvaccinated population (<60yo M = 0.08; ≥60yo M = 0.3). Based on IgG levels, AstraZeneca was significantly more effective than CoronaVac in the fully immunized population (p = 0.0001) or in individuals below (p = 0.0136) and above (p = 0.0001) 60yo. This fact could be explained by the time after full immunization of individuals and their age as differences were observed between CoronaVac (M = 10 weeks, M = 70yo) and AstraZeneca (M = 4 weeks, M = 60yo) (see Fig 4 for details). Overall, vaccinee age impacted IgG levels measured by

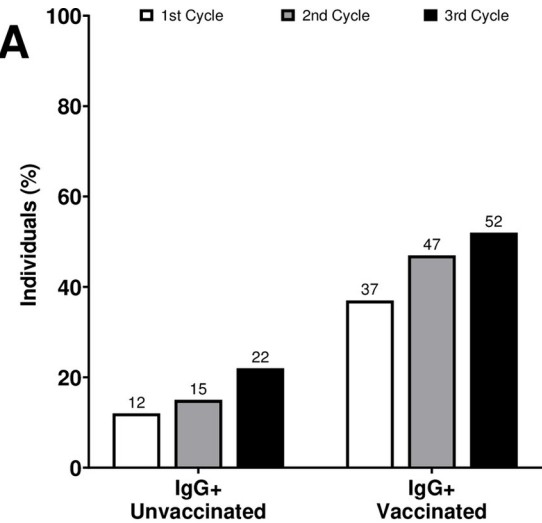

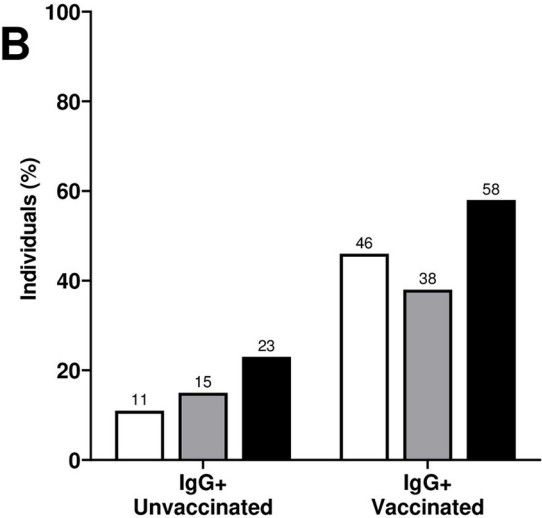

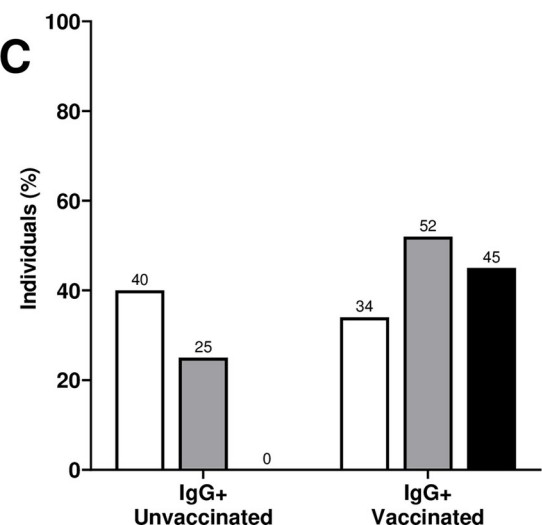

**Fig 2. Percentage of anti-SCOV2 immunoglobulin positivity in each study cycle.** A, B and C) Anti-SCOV2
Immunoglobulin profile of unvaccinated participants. A) All vaccinated participants. B) <60 years old group. C) ≥60
years old group. White bars represent the 1st cycle, grey bars the 2nd cycle and black bars the 3rd cycle.

CLIA assays. Individuals older than 60yo showed lower IgG levels compared to younger age
groups (<60yo).

When RT-PCR positive results were analyzed in vaccinated and unvaccinated groups, a
clear difference in IgG levels was observed. Most of RT-PCR+ samples had lower IgG titers
(n = 14) (Fig 3, red dots).

We further investigated neutralizing antibodies (NAb) in a selected group of IgG positive
individuals with the Lumit assay (Fig 3D–3F). We found that 79% of IgG+ participants vacci-
nated with AstraZeneca developed NAb in a relevant title (>70%). On the other hand, Coro-
naVac induced NAb in 24% of total IgG+ individuals. In contrast, only 10% of unvaccinated
IgG+ participants had NAb and this was significantly lower compared to CoronaVac fully vac-
cinated group ($\chi2$ = 6.9, p = 0.008403). There was a clear association between the level of anti-
SCOV2 IgG measured by CLIA and the percentage of individuals carrying positive levels of
NAb in our study. When anti-SCOV2 antibody levels were breakdown into three OD/Cut-off
windows (1 to 5; 5 to 10; and beyond 10) we found 41, 77, and 96% of individuals showing
detectable levels of NAb, respectively. Interestingly, we found that the majority of RT-PCR
+ individuals presented high levels of NAb (n = 7) with only three showing low NAb levels
(<70% of RDB:ACE2 inhibition) regardless the vaccination status.

A correlation between NAb production and anti-SCOV2 IgG levels in AstraZeneca fully
immunized subjects (Fig 3D–3F) could also be found. Nearly 100% of individuals of this group
showed significantly higher IgG levels when compared to IgG+ unvaccinated population,
regardless age (global, <60 and >60yo; p = 0.0001). In comparison to CoronaVac, AstraZe-
neca elicited more NAb production in individuals above 60yo (p = 0.0001) as well as in overall
fully immunized ones (p = 0.0001). We did not see any statistical difference in NAb levels
between fully immunized CoronaVac and unvaccinated IgG+ individuals. Although there was
a small number of individuals vaccinated with Pfizer and Janssen vaccines, their effectiveness
in terms of production of anti-SCOV2 IgG and NAb was also analyzed. Janssen (n = 6) fully
immunized individuals had the highest anti-SCOV2 IgG levels (M = 13.36) when compared to
the AstraZeneca fully immunized group (M = 4,19). Although we did not find Pfizer fully
immunized individuals in our study, participants who received one jab of Pfizer (n = 68) pro-
duced strong levels of anti-SCOV2 IgG (M = 8.02) and NAb (M = 99%). We did not observe
RT-PCR+ subjects vaccinated with Pfizer or Janssen. However, the high IgG titers of Pfizer
and Janssen vaccinees could reflect their recent immunization (<2 months). CoronaVac fully
immunized individuals had an average time after the second dose of 10 weeks (range 2–24),
whereas for AstraZeneca fully immunized individuals this was 4 weeks (range 2–17) (Fig 4A).
CoronaVac fully immunized individuals had an average age of 70yo, whereas AstraZeneca
fully immunized individuals had an average age of 60yo (Fig 4B). Participants vaccinated with
Pfizer and Janssen had their immunizations recently given–Pfizer 1st dose 3 weeks (range
0–10); Janssen 3 weeks (range 2–4). Moreover, those participants were younger (M = 47yo)
than individuals fully immunized with CoronaVac and AstraZeneca vaccines.

Besides our limited RT-PCR+ samples, we could find a level of protection against SARS-
CoV-2 infection between vaccinated and unvaccinated population in our study (34%).
However, if we stratify individuals fully vaccinated according to vaccine kind, CoronaVac vac-
cinated subjects presented no level of protection contrasting to the AstraZeneca fully immu-
nized ones. Moreover, when incidence data and immunization rate were combined for each
round of the study, an inverse correlation is found (Fig 5). As immunization rates increase, the

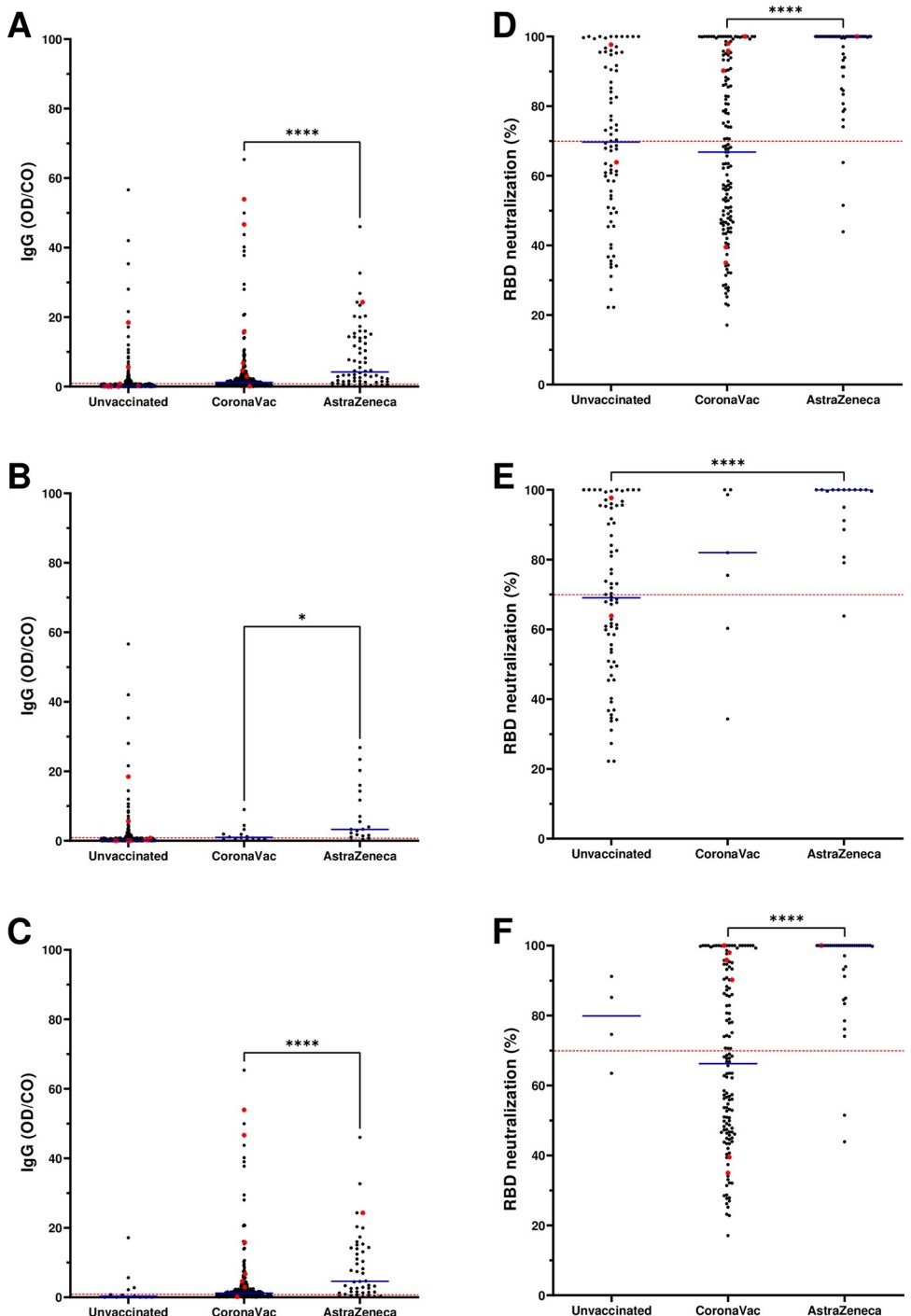

**Fig 3. Comparisons among unvaccinated and fully immunized groups according to their anti-SCOV2 IgG and NAb levels.** A, B and C) Anti-SCOV2 IgG serum levels according to age (overall, <60 and ≥60 years old, respectively). D, E and F) NAb serum levels of overall, <60 and ≥60 years old groups, respectively. Red line represents the cut offs (≥1.0 and ≥70%). Blue lines stands for the median of each group. Red dots represent SARS-CoV-2 RT-PCR positive individuals in each group. * p value <0.05; **** p value <0.0001.

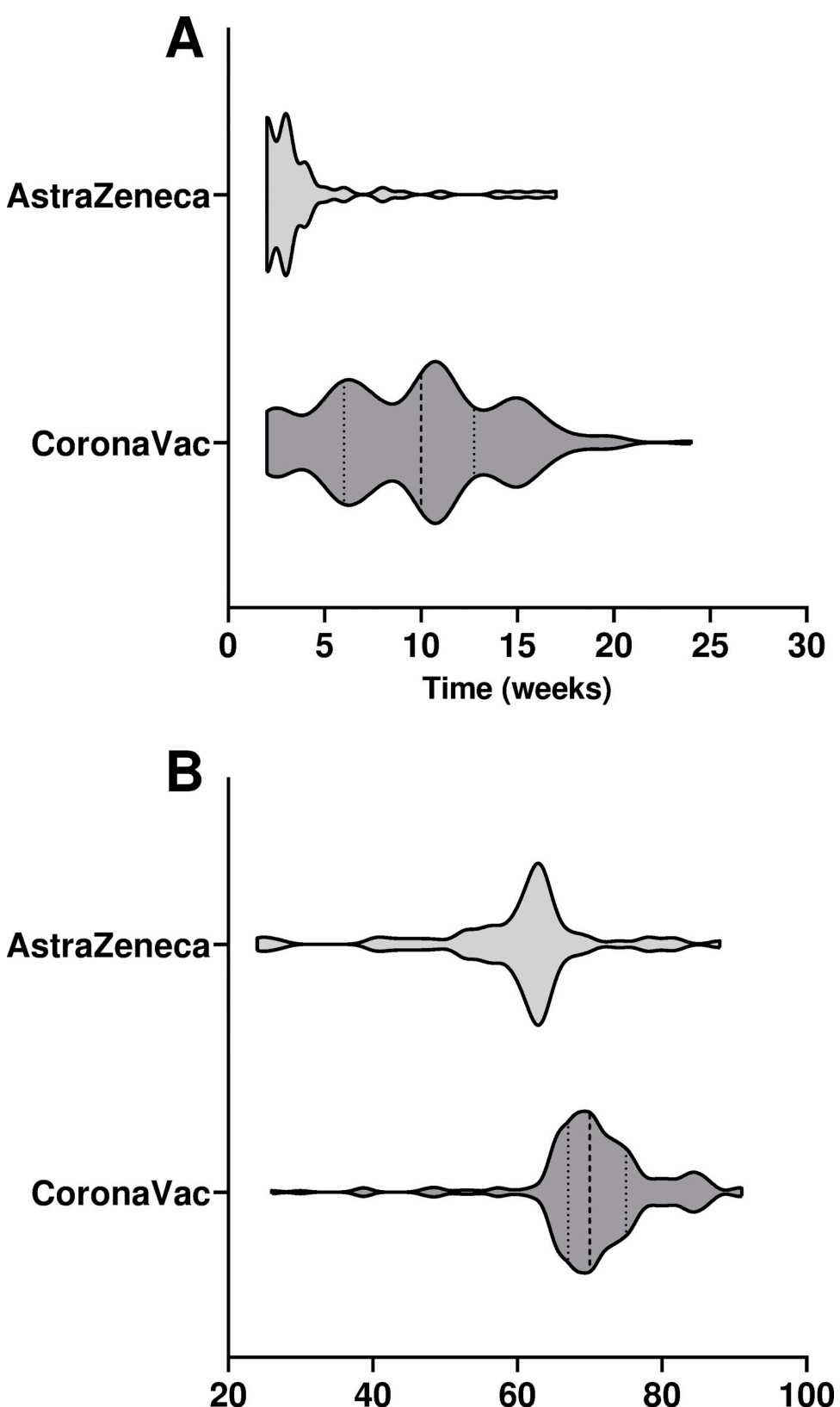

**Fig 4. Fully vaccinated groups distribution.** A) Distribution of CoronaVac and AstraZeneca fully vaccinated groups according to time after the end of immunization scheme. B) Age distribution of CoronaVac and AstraZeneca fully vaccinated groups.

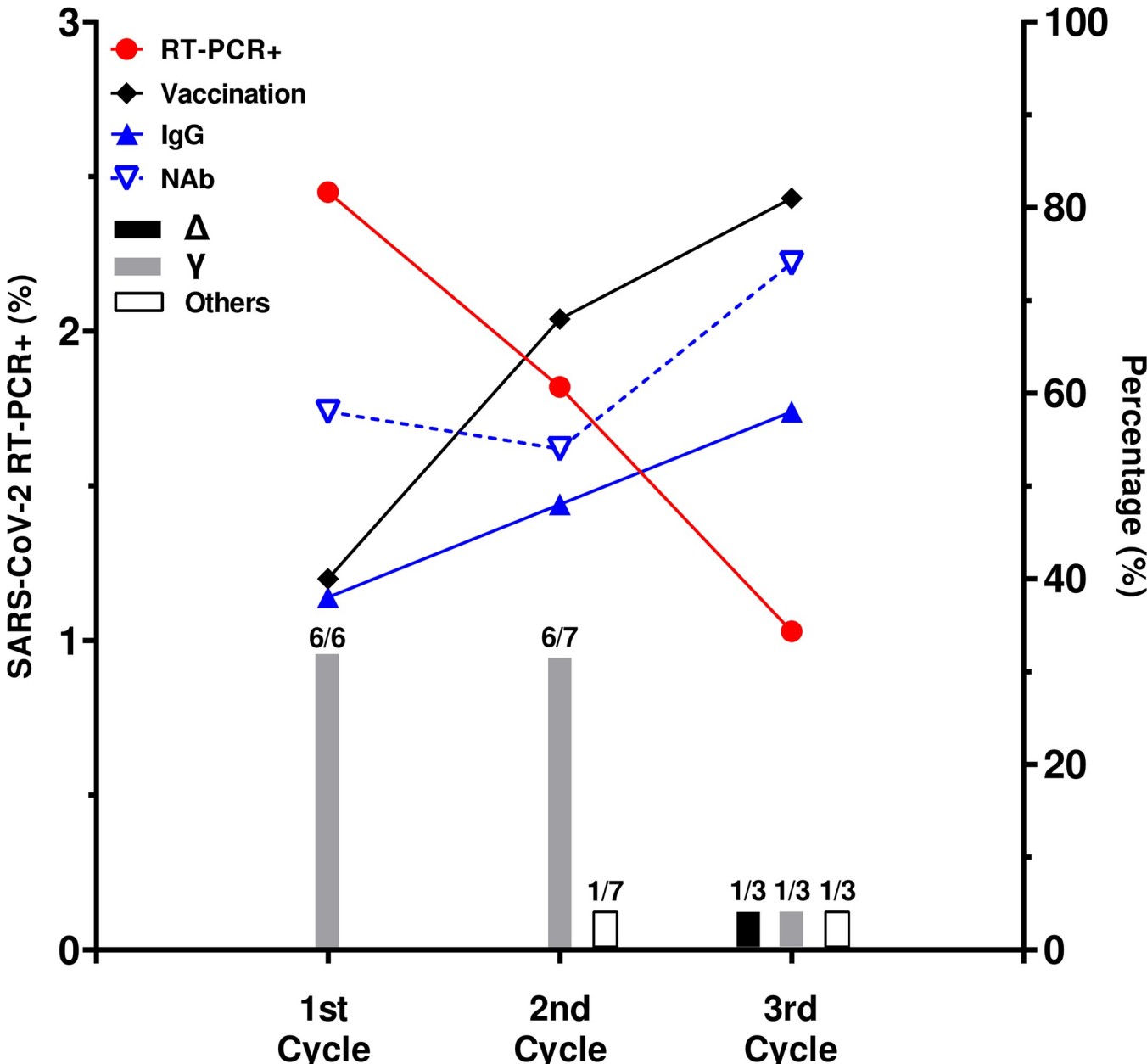

**Fig 5. Impact of immunization on COVID-19 incidence on the studied population.** On the left Y axis: red line shows de COVID-19 incidence on the studied population over the three round. On the right Y axis: 1) black line represents the percentage of vaccinated participants through the three rounds; 2) blue line shows the percentage of SCOV2 IgG+ individuals (IgG OD/CO >1.0); 3) dotted blue line points the percentage of SCOV2 IgG+ individuals carrying detectable levels of NAb. Black, grey, and white bars represent the frequency of Delta (Δ), Gamma (γ) and other VOCs, respectively, in the city of Maricá when all three rounds were performed.

number of individuals showing detectable levels of IgG anti-SARS COV2 as well as detectable NAb over the cycles increases at the same pace. Contrasting to that, COVID-19 incidence measured by RT-PCR dropped drastically (Fig 5). In addition, if the proportion of VOCs presented in each cycle are compared with COVID-19 incidence, a drop in incidence is noticed regardless to a shift of VOCs proportion (Fig 5). At the beginning of our study Gamma variant was the most frequent (90%) and was substituted by Delta variant in the last study round.

## Discussion

Population-based data on COVID-19 are essential for guiding policies and evaluating public health interventions made during pandemics [13–16]. So far, there are few such studies, particularly from lower or middle-income countries [15, 17]. Our study captures epidemiological data from individuals randomly selected in three districts of Maricá, Brazil. We selected 39 urban census tracts with probability proportional to size sampling in three sentinel round, collecting data and clinical specimens of 384 individuals in each round. The data presented here corroborate previous knowledge that the presence of infected individuals in the same house is a major risk for SARS-CoV-2 infection worldwide [17, 22–26]. In fact, we observed a prevalence of RT-PCR+ participants and the presence of a household with COVID in our study. After sociodemographical analysis, we could find an association between educational level and SARS-CoV-2 RT-PCR positivity. It is well known that COVID-19 has a higher incidences among individuals with lower levels of education [15, 17, 22–24, 27–30]. Although we could not point statistically differences, we also found a high number of RT-PCR+ individuals having running nose, cough, and headache. As seen by others, it was difficult to establish a specific group of symptoms related to SARS-CoV-2 infection [31].

The global rate of RT-PCR+ individuals in the study was 1.76%. We observed a progressive reduction on RT-PCR+ cases throughout the study rounds. Fifty five percent of RT-PCR+ participants received at least one vaccine jab and 40% were fully immunized with CoronaVac or AstraZeneca vaccines. Most of RT-PCR+ participants reported recent symptoms related to COVID-19. Approximately half of the RT-PCR+ symptomatic participants were vaccinated, and we found no significant differences in N1 Ct values between vaccinated and unvaccinated individuals. This indicates that the vaccine itself might not impact viral load during acute infections. This could be due to the kind of VOC in those infected individuals [32, 33].

The global vaccination rate observed in our study was 65% in participants receiving at least one vaccine jab, and CoronaVac and AstraZeneca were the most frequent vaccines used. We observed a sustained increase in anti-SCOV2 IgG positive participants over the three rounds of the study. The overall anti-SCOV2 IgG positive individuals rate was 48%. In comparison to unvaccinated participants, the anti-SCOV2 IgG positivity in vaccinated individuals was nearly three times higher. However, we observed a significant increase in the amount of unvaccinated IgG positive individuals in the 3rd round, which matched with the increase of Delta variant in Rio de Janeiro State and Maricá [8]. Most of them reported no COVID-19 symptoms in the last 30 days prior to interview, suggesting an asymptomatic infection in this group. This number contrasts with the RT-PCR+ data, where more symptomatic infections were observed. This fact could be due to the introduction of Delta variant, known to be more transmissible and previously related to asymptomatic infections when compared to Gamma [28, 30–32, 34, 35].

The most frequent vaccines received by our population were CoronaVac and AstraZeneca. Both vaccines produced significant levels of anti-SCOV2 IgG in fully immunized individuals when compared to unvaccinated ones. Based on IgG levels, AstraZeneca was significantly more effective than CoronaVac in fully immunized individuals. This could be explained by the

immunization strategy adopted in Brazil [12], since the COVID-19 National Immunization Program started with CoronaVac immunization in elderlies with AstraZeneca and other vaccines (Pfizer and Janssen) coming right after that in adult immunization. In fact, in this study, participants vaccinated with AstraZeneca were younger and had less time after full immunization when compared with CoronaVac vaccinees. Another observation was that vaccinee age impacted the level of IgG. Vaccinated individuals older than 60yo had lower IgG levels when compared to a younger group. As previously demonstrated by several studies, this could represent the basis by which a 3[rd] dose was rapidly recommended in elderly across many countries [36–38].

Our study showed a clear association between anti-SCOV2 IgG levels and the percentage of individuals with detectable levels of NAb. CoronaVac and AstraZeneca produced significant levels of NAb in anti-SCOV2 IgG positive vaccinated individuals. Again, AstraZeneca was significantly more effective in NAb production than CoronaVac considering the fully immunized population. This could be explained by participants age and/or the long time after the second CoronaVac dose [38, 39]. Interestingly, most of the individuals that showed RT-PCR+ in the vaccinated group had a strong neutralization title indicating a fast NAb production after SARS-CoV-2 infection [40]. Only a few percentages of unvaccinated anti-SCOV2 IgG positive participants had NAb. Some of them with high anti-SCOV2 IgG and Nab levels, that could indicate infection close to each study round [41, 42].

In contrast to several studies [43–45], we could see a small level of protection against SARS-CoV-2 infection in vaccinated population in our study, besides our limited sample size (34%). The same level of protection was seen in AstraZeneca fully vaccinated individuals. In contrast, we could not find any level of protection in CoronaVac fully immunized subjects. However, it was possible to see an inverse correlation between incidence data and immunization rate. As the number of individuals showing detectable levels of IgG anti-SARS COV2 and NAb increased, incidence of RT-PCR+ dropped drastically.

Nonetheless, COVID-19 incidence drop should not only be interpreted in the light of vaccination status. Community transmission rates in a specific period and mitigation measurements must be considered. We found no statistical correlations on our non-pharmacological measures and social distance compliance analysis. It will be necessary, in future studies, to increase the number of and/or the time among rounds to cover different periods of community transmission.

In addition, Maricá epidemiological data such as severe case hospitalization as well as mortality has dropped 3 times in the same period, corroborating our study findings [7]. Of note, we observed a shift of VOCs across the three cycles of our study. Indeed, there was a shift of VOCs in the Rio de Janeiro State [8]. At the beginning of our study Gamma variant was the most frequent (90%) in the population whereas Delta variant appeared only in six percent of the cases. At the end of study Delta variant was found in 90% of individuals studied in a SARS-CoV-2 VOC sampling done in Rio de Janeiro State [8]. Then, we can argue that this drop in incidence at the same time of Delta VOC introduction could be due to a high vaccination rate.

## Conclusion

Our findings show that the number of individuals carrying detectable anti-SCOV2 IgG and NAb levels was bigger in vaccinated compared to unvaccinated groups, proving the importance of the vaccination to attain noticeable levels of herd immunity against SARSCoV-2. We found a decreased incidence of COVID-19 throughout the study, and this was correlated with vaccination status, IgG levels and NAb titles across study rounds. Our data clearly support the

use of vaccines to drop the incidence of SARS-CoV-2 infection and the consequent reduction in morbidity and mortality associated with COVID-19. We also found a drop in the anti-SCOV2 IgG levels as well as Nab titers in individuals vaccinated with CoronaVac at more than 10 weeks. We could not see these drops in the AstraZeneca, Pfizer and Janssen vaccines, probably to a short period of time after immunization until sampling. This kind of sampling methodology is an inexpensive way to monitor the spread of COVID-19 in a population and to evaluate the impact of vaccination in low-income countries.

## Supporting information

**S1 Table. Detailed sociodemographic, non-pharmacological measures and clinical data from participants.**
(PDF)

**S1 Data.**
(XLSX)

## Author Contributions

**Conceptualization:** Carlos Alberto de Senna Costa, Marcelo Costa Velho Mendes de Azevedo, Mônica Barcellos Arruda, Patricia Alvarez, Orlando Costa Ferreira, Jr, Alberto Carlos Melo de Almeida, Amilcar Tanuri.

**Data curation:** Thiago Silva Frauches, Mônica Barcellos Arruda, Orlando Costa Ferreira, Jr, Aline Santos, Alberto Carlos Melo de Almeida.

**Formal analysis:** Thiago Silva Frauches, Orlando Costa Ferreira, Jr.

**Funding acquisition:** Carlos Alberto de Senna Costa, Celso Pansera.

**Investigation:** Claudia dos Santos Rodrigues, Marcelo Costa Velho Mendes de Azevedo, Hanna Beatriz Vieira da Silva Ramos, Wilson Rodrigues de Souza Junior, Andréa Ribeiro Costa, Andreia Edwirges de São Carlos, Patricia Alvarez.

**Methodology:** Claudia dos Santos Rodrigues, Marcelo Costa Velho Mendes de Azevedo, Hanna Beatriz Vieira da Silva Ramos, Wilson Rodrigues de Souza Junior, Andréa Ribeiro Costa, Adriana Cardoso Camargo, Adriana Halfeld Alonso, Fábio Álvaro dos Santos, Hércules da Silva Oliveira, Janaína Guimarães Coelho, Joyce Florentina da Silva Sobral, Luciane Cardoso dos Santos Rodrigues, Marcio Martins Casaes Ferreira, Patricia Laureano, Raquel Adalgiza da Paz Fernandes, Renata da Silva Santos, Rose Mary Carvalho dos Santos, Sanderson Milagres, Vanessa Cristina Conceição dos Santos, Jussara Teixeira Silva, Tatiana Martins da Silva, Malu Gabriela Costa da Rocha, Amorim Mourão de Araújo Ramos, Fernanda Martins de Almeida Bastos, Daina Raylle Francisco, Sabrina dos Santos Rosa, Layla Corrêa Linhares, Raissa Rodrigues Organista, Leandro Bastos, Maria Magdalena Kelly Pinto, Jean Pablo Lima do Nascimento, João Pedro Moura da Silveira, Mateus Quintanilha dos Santos, Nathaly Santos da Silva, Nayra Cristina dos Santos Ferreira, Rafael Brito Ramirez Reis, Ruan Fonseca de Oliveira, Valdinei de Oliveira Sá, Thyago Ramos de Siqueira Hammes, Juliano de Oliveira Monteiro, Pedro Henrique Cardoso, Patricia Alvarez, Richard Araujo Maia, Liane de Jesus Ribeiro, Aline Santos.

**Project administration:** Carlos Alberto de Senna Costa, Lauro Garcia, Celso Pansera.

**Resources:** Celso Pansera.

**Supervision:** Carlos Alberto de Senna Costa, Marcelo Costa Velho Mendes de Azevedo, Michelle de Moraes Ferreira, Hanna Beatriz Vieira da Silva Ramos, Aline Santos, Alberto Carlos Melo de Almeida, Celso Pansera.

**Validation:** Pedro Henrique Cardoso, Mônica Barcellos Arruda, Patricia Alvarez, Orlando Costa Ferreira, Jr, Alberto Carlos Melo de Almeida.

**Visualization:** Marcelo Costa Velho Mendes de Azevedo, Pedro Henrique Cardoso, Mônica Barcellos Arruda, Patricia Alvarez, Orlando Costa Ferreira, Jr, Alberto Carlos Melo de Almeida, Celso Pansera.

**Writing – original draft:** Thiago Silva Frauches, Carlos Alberto de Senna Costa, Amilcar Tanuri.

**Writing – review & editing:** Thiago Silva Frauches, Amilcar Tanuri.

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
