## [Decision Letter · Decision Letter 0]

22 Mar 2022

PONE-D-22-03333COVID-19 population-based survey in Maricá, Brazil: the impact of vaccination in viral incidence.PLOS ONE

Dear Dr. Tanuri,

Thank you for submitting your manuscript to PLOS ONE. After careful consideration, we feel that it has merit but does not fully meet PLOS ONE’s publication criteria as it currently stands. Therefore, we invite you to submit a revised version of the manuscript that addresses the points raised by both reviewers.

We look forward to receiving your revised manuscript.

Kind regards,

Odir Antonio Dellagostin

Academic Editor

PLOS ONE

Journal Requirements:

Reviewers' comments:

Reviewer's Responses to Questions

**Comments to the Author**

1. Is the manuscript technically sound, and do the data support the conclusions?

Reviewer #1: Yes

Reviewer #2: Partly

2. Has the statistical analysis been performed appropriately and rigorously? 

Reviewer #1: Yes

Reviewer #2: No

3. Have the authors made all data underlying the findings in their manuscript fully available?

Reviewer #1: Yes

Reviewer #2: Yes

4. Is the manuscript presented in an intelligible fashion and written in standard English?

Reviewer #1: Yes

Reviewer #2: No

5. Review Comments to the Author

Reviewer #1: The manuscript “COVID-19 population-based survey in Maricá, Brazil: the impact of vaccination in viral incidence” submitted to PlosOne Journal presented data about the decrease in the incidence of COVID-19 according to the level of vaccination in a Brazilian city, investigating data of three repeated cross-sectional from 1134 subjects in a population-based epidemiological study from May to August 2021.

It is an interesting study. However, I would like to discuss some aspects with the authors to improve the manuscript before publication. I recommend the authors revise the results description and indications of the figures and tables in the text. Also, a revision of the conclusion. In summary, the study can be published after minor revision. Questions, comments, and suggestions are listed below:

Major comments:

#1 Introduction: Since it is currently well known that vaccination prevents COVID-19 infection, I recommend including the reasons to explore this issue in this article. Some points:

For example, in comparison or sequence with other studies, what novelty this relevant study may corroborate? Is it because of the Brazilian context during the period? Is it because of the specific behavior of the population in this local? Was it not expected that the vaccine promotes an increase of neutralizing antibody levels, or during this period, this was unknown information? The type of vaccine applied vs. variant prevalence can be included as a novel aspect for readers?

To summarize, it is not clear the novelty end contribution of the study for readers in the introduction section. The authors presented clearly the study objective in the introduction: “our aim is to investigate SARS-CoV-2 antibody (anti-SCOV2) prevalence by city and RT-PCR results in a repeated cross-sectional study in a municipality”. However, the lack of information in the field that can be answered with the study is not completely clear in the introduction. For example, there are some comparisons made about vaccine type in the study. After reading the introduction section, this type of comparison is not expected for readers.

#2 Methods: The authors informed: “Subjects below 18 years old were excluded, and if the selected individual did not provide a sample, another household member was randomly selected.” During the period, teenagers had access to vaccination in this place? If they don’t, I suggest including this information as a reason not to include < 18 yo subjects. If they had access to the vaccine, please inform readers why not include them. All other information was adequately presented in methods such as ethical approval, population-based sampling procedures (areas and subjects), sample collection and analytical procedures (RT-PCR, serological tests…),

#3 Figure 1B showed an increase in the N1 target CT during the 2° transversal data collection cycle and a decrease in the next cycle. Is there some special meaning about this result according to the variant/period to be interpreted and discussed in the manuscript?

#4 The description of the results presented in figure 2 needs a revision. There is some repetition of information. Also, the text informed (line 292): “On the third cycle the anti-SCOV2 IgG positive rate decreased in vaccinated individuals with age above 60yo (Fig 2C)”. However, the decreasing is showed in the figure of unvaccinated individuals in the figure. (is it possible that groups were presented in the figure changed?). Also, the legend of figure 2 includes figures 2D, 2E, and 2F. However, these figures were not presented. In another hand, the results about the prevalence of NAB+ in unvaccinated vs. vaccinated subjects showed in the figure 2 were not described in the text.

#5 The discussion is adequately centered on the effect of vaccines on the reduction of COVID-19 impact. Also, there is a discussion about the type of vaccine. Because of these, I recommend above to include this expectation in the introduction for readers. Also, since the vaccine is not only the way to avoid the incidence of COVID-19, I recommend authors include a short paragraph in the discussion about the local recommendations and adherence of subjects in the period of the study in terms of social distancing and use of masks, for example. This is necessary to better understand of the context and also may help to improve the support of the conclusion: “The effect of vaccines”.

#6 Conclusion: I recommend deleting the text between lines 531-535, it is a description of the study again. Also, authors concluded an additional point not expected: “We also found a drop in the anti-SCOV2 IgG levels as well as NAb in individuals vaccinated CoronaVac more than 10 weeks supporting the use of an additional vaccine dose to boost the immune response in these individuals.” However, the study showed that Coronavac presented lower levels of protection in comparison with AstraZeneca and argued that these findings were due to the time after completing the vaccination scheme. For supporting this conclusion adequately, It was necessary to include results about the analysis of protection levels vs. time after vaccination in both vaccine regimens. However, the authors agree that this is not possible since “We could not see these drops in the AstraZeneca, Pfizer and Janssen vaccines due to a short period of time after immunization before our sampling”. Thus, since the decrease in the protection was not adequately presented in the study, this conclusion may not be adequate. I would like to hear the authors about it.

Minor points:

#7 Authors described the result of 26% of family history of COVID-19 as follows: “almost one-third of the 211 interviewed participants reported disease in relatives cohabiting the household.”. I suggest that the authors change this text for approximately one-fourth, not one-third.

#8 After the description of the results on lines 223-229, please indicate for readers to find these results in table S1;

#9 Is the text described in lines 348-351 about the results presented in figure 3? The comparison about IGG levels using only COVID positive subjects could be presented with more details, in the text of including a graph in figure 3.

Reviewer #2: The study evaluates the seropositive gains CPVO-19 according with vaccinal status, over time in Marica-RJ.

1. I suggest that the authors followed the STROBE statement Statement—Checklist of items that should be included in reports of cross-sectional studies.

2. The title must contain the study design.

3. The title specifies that the study evaluates the impact of the vaccination in viral incidence, but in the abstract the objectives was to evaluate the immunological status (antibodies against SARS-COV2), not the viral incidence (see title, pg 57 and pg 105-107). Please adapt the title.

4. The conclusion of the abstract refers to drop in mortality, that was not supported in the data presented. There is no mortality data in the study.

5. In the methods section, the sampling process description should include the period of recruitment and also explain how the study size was arrived at. The eligibility criteria and exclusion must be described. How the study handled with illiterate participants or individual with special needs?

6. The main outcomes need to be clearly defined, as well as exposures, predictors, potential confounders, and effect modifiers.

7. The data analysis must contain the test used do calculate effectiveness and also include how the immunological variables were handled. Additionally, must contain the software used for data analysis.

8. Regarding the results, please confirm that the number of participants with previous COVID was the same (13) in all three cycles – Table 1.

9. Please include a measure of difference in the S1 table and Table 1 or add the measurements to the text. The paragraph 223-229 stated that there are differences between groups, but there is no statistical test associated to that.

10. In line 275 the authors stated that there is an increase in % of participants, but there is no statistical test associated. Please include a statistical test in Table 3.

11. The Figure 2 need a legend for the abbreviations - NAb+.

12. Please correct the figure 3 and add p-value for the differences between groups or add a legend to state what the number 1 1 1 1 means.

13. Please specify which figure/table the results from lines 348-351 refers to sentence 360-362. As well as for the correlation states in lines 371-372.

14. In Figure 5 is very confuse. The Y axis goes from 0-3 and the secondary axes from 0-100% without axis specification.

15. Inline 401, is not clear with outcome the effectiveness analysis refers to. Please clarify and add the statistical test.

16. The authors make different statements regarding the association of risk factors and the PCR positivity. These affirmation needs to be based on multivariate analysis to avoid confounding bias. Please add a multivariate analysis to evaluate the risk factors (Pgs 438-448) or modify the paragraph.

17. The authors should include a paragraph with the weaknesses of the study.

18. The incidence of COVID-19 could be not only because of vaccination status but also due to the probability of infection in a specific period of time, that depend also from community transmission rates and mitigation measurements. Please discuss this.

19. The manuscript needs a grammar English review.

6. PLOS authors have the option to publish the peer review history of their article (what does this mean?). If published, this will include your full peer review and any attached files.

Reviewer #1: **Yes: **Thiago Gomes Heck

Reviewer #2: No

---

## [Author Response · Author response to Decision Letter 0]

23 Apr 2022

To PlosOne 

REF: Rebuttal Letter Manuscript PONE-D-22-03333.

 Rio de Janeiro, 3/30/2022

Dear Editor 

Thank you for the excellent review done in our manuscript entitled “COVID-19 population-based survey in Maricá, Brazil: the impact of vaccination in viral incidence.” I am sure that reviewers´ comments will make the article clearer and more interesting for readers. 

Please find bellow our response to reviewers queries. 

Reviewer #1: 

Major comments:

#1 Introduction: Since it is currently well known that vaccination prevents COVID-19 infection, I recommend including the reasons to explore this issue in this article. Some points:

For example, in comparison or sequence with other studies, what novelty this relevant study may corroborate? Is it because of the Brazilian context during the period? Is it because of the specific behavior of the population in this local? Was it not expected that the vaccine promotes an increase of neutralizing antibody levels, or during this period, this was unknown information? The type of vaccine applied vs. variant prevalence can be included as a novel aspect for readers?

To summarize, it is not clear the novelty end contribution of the study for readers in the introduction section. The authors presented clearly the study objective in the introduction: “our aim is to investigate SARS-CoV-2 antibody (anti-SCOV2) prevalence by city and RT-PCR results in a repeated cross-sectional study in a municipality”. However, the lack of information in the field that can be answered with the study is not completely clear in the introduction. For example, there are some comparisons made about vaccine type in the study. After reading the introduction section, this type of comparison is not expected for readers.

We concur with reviewer comment and we have included a new text and three new references (9-11) to discuss the impact of vaccination in viral incidence. 

#2 Methods: The authors informed: “Subjects below 18 years old were excluded, and if the selected individual did not provide a sample, another household member was randomly selected.” During the period, teenagers had access to vaccination in this place? If they don’t, I suggest including this information as a reason not to include < 18 yo subjects. If they had access to the vaccine, please inform readers why not include them. All other information was adequately presented in methods such as ethical approval, population-based sampling procedures (areas and subjects), sample collection and analytical procedures (RT-PCR, serological tests…),

The reason we included only adults (>18 years old by Brazilian laws) is that collecting samples from childrens (<18 years old) needs the approval/consent of the kid guardian and it would complicate our study. In addition, our IRB has very restricted rules to include children in epi studies. 

#3 Figure 1B showed an increase in the N1 target CT during the 2° transversal data collection cycle and a decrease in the next cycle. Is there some special meaning about this result according to the variant/period to be interpreted and discussed in the manuscript?

We agree and we have included a new paragraph discussing this issue in the discussion section. 

#4 The description of the results presented in figure 2 needs a revision. There is some repetition of information. Also, the text informed (line 292): “On the third cycle the anti-SCOV2 IgG positive rate decreased in vaccinated individuals with age above 60yo (Fig 2C)”. However, the decreasing is showed in the figure of unvaccinated individuals in the figure. (is it possible that groups were presented in the figure changed?). Also, the legend of figure 2 includes figures 2D, 2E, and 2F. However, these figures were not presented. In another hand, the results about the prevalence of NAB+ in unvaccinated vs. vaccinated subjects showed in the figure 2 were not described in the text.

This was corrected in the new version of the manuscript (see corrected copy in annex). 

#5 The discussion is adequately centered on the effect of vaccines on the reduction of COVID-19 impact. Also, there is a discussion about the type of vaccine. Because of these, I recommend above to include this expectation in the introduction for readers. Also, since the vaccine is not only the way to avoid the incidence of COVID-19, I recommend authors include a short paragraph in the discussion about the local recommendations and adherence of subjects in the period of the study in terms of social distancing and use of masks, for example. This is necessary to better understand of the context and also may help to improve the support of the conclusion: “The effect of vaccines”.

Thanks for this remind and we have included a new paragraph to discuss this point in Conclusion section. 

#6 Conclusion: I recommend deleting the text between lines 531-535, it is a description of the study again. Also, authors concluded an additional point not expected: “We also found a drop in the anti-SCOV2 IgG levels as well as NAb in individuals vaccinated CoronaVac more than 10 weeks supporting the use of an additional vaccine dose to boost the immune response in these individuals.” However, the study showed that Coronavac presented lower levels of protection in comparison with AstraZeneca and argued that these findings were due to the time after completing the vaccination scheme. For supporting this conclusion adequately, It was necessary to include results about the analysis of protection levels vs. time after vaccination in both vaccine regimens. However, the authors agree that this is not possible since “We could not see these drops in the AstraZeneca, Pfizer and Janssen vaccines due to a short period of time after immunization before our sampling”. Thus, since the decrease in the protection was not adequately presented in the study, this conclusion may not be adequate. I would like to hear the authors about it.

We agree and we modified this paragraph in the Discussion section removing this comparison. 

Minor points:

#7 Authors described the result of 26% of family history of COVID-19 as follows: “almost one-third of the 211 interviewed participants reported disease in relatives cohabiting the household.”. I suggest that the authors change this text for approximately one-fourth, not one-third.

Thanks for your suggestion, the change was done.

#8 After the description of the results on lines 223-229, please indicate for readers to find these results in table S1;

We included the table S1 indication in the text.

#9 Is the text described in lines 348-351 about the results presented in figure 3? The comparison about IGG levels using only COVID positive subjects could be presented with more details, in the text of including a graph in figure 3.

You are correct, the paragraph describes results presented in figure 3. We included the indication in the text and pointed those red dots in the figures 3 represent COVID positive subjects.

Reviewer #2: The study evaluates the seropositive gains CPVO-19 according with vaccinal status, over time in Marica-RJ.

1. I suggest that the authors followed the STROBE statement Statement—Checklist of items that should be included in reports of cross-sectional studies.

We appreciate all the suggestions, and we checked the items in the STROBE statement and placed inside the manuscript. 

2. The title must contain the study design.

The study design was included in the title. 

3. The title specifies that the study evaluates the impact of the vaccination in viral incidence, but in the abstract the objectives was to evaluate the immunological status (antibodies against SARS-COV2), not the viral incidence (see title, pg 57 and pg 105-107). Please adapt the title.

We agree with your point. Title was adapted. 

4. The conclusion of the abstract refers to drop in mortality, that was not supported in the data presented. There is no mortality data in the study.

We agree with your point. The results presented on this study do not support the drop of mortality. That affirmation was excluded in the text. 

5. In the methods section, the sampling process description should include the period of recruitment and also explain how the study size was arrived at. The eligibility criteria and exclusion must be described. How the study handled with illiterate participants or individual with special needs?

The period was included in the sampling strategy. The exclusion criteria were improved. For illiterate participants, we included a detail in data specimen collection, the questionnaire was applied by interviewers using an application. We did not use printed questionnaire in the study.

6. The main outcomes need to be clearly defined, as well as exposures, predictors, potential confounders, and effect modifiers.

We have included a new paragraph in Discussion section to place some of the cofounders and modified effects of our study.

7. The data analysis must contain the test used do calculate effectiveness and also include how the immunological variables were handled. Additionally, must contain the software used for data analysis.

Effectiveness calculations were explained at lines 207-210. We mentioned the software at line 206.

8. Regarding the results, please confirm that the number of participants with previous COVID was the same (13) in all three cycles – Table 1.

The percentage of participants with previous COVID was the same in all three cycles, but the number differed in each cycle. The number of participants with previous COVID are in parenthesis. We had to change the table layout to fit the page. A landscape layout is the best choice to present this table.

9. Please include a measure of difference in the S1 table and Table 1 or add the measurements to the text. The paragraph 223-229 stated that there are differences between groups, but there is no statistical test associated to that.

In this paragraph our intention was only to describe the different numbers of individuals with specific characteristics not a formal statical analysis. To make our text more clear we placed S1 Table reference at the end of this paragraph. Then the readers can find these number direct in Table S1. 

10. In line 275 the authors stated that there is an increase in % of participants, but there is no statistical test associated. Please include a statistical test in Table 3.

The statistical analysis was included.

11. The Figure 2 need a legend for the abbreviations - NAb+.

The data about neutralizing antibodies were not supposed to be present in figure 2. Those data were excluded.

12. Please correct the figure 3 and add p-value for the differences between groups or add a legend to state what the number 1 1 1 1 means.

Sorry for that. The figured lost its configuration on the exportation process. The figure was fixed.

13. Please specify which figure/table the results from lines 348-351 refers to sentence 360-362. As well as for the correlation states in lines 371-372.

The lines 348-351 refer to figure 3. The indication was added in the text. The sentences 360-362 we meant association. We excluded the term correlation.

14. In Figure 5 is very confuse. The Y axis goes from 0-3 and the secondary axes from 0-100% without axis specification.

We also had a loss of configuration in this figure, it was fixed. But the axis meaning are detailed in the legends. Both show percentage values. We chose different scales for aesthetics proposal to try to demonstrate the drop on incidence (left Y axis) while increment of vaccination and Ig (right Y axis)

15. Inline 401, is not clear with outcome the effectiveness analysis refers to. Please clarify and add the statistical test.

The effectiveness refers to all vaccines combined in participants who received at least one shot.

16. The authors make different statements regarding the association of risk factors and the PCR positivity. These affirmation needs to be based on multivariate analysis to avoid confounding bias. Please add a multivariate analysis to evaluate the risk factors (Pgs 438-448) or modify the paragraph.

We agreed and changed the paragraph.

17. The authors should include a paragraph with the weaknesses of the study.

See comment 18.

18. The incidence of COVID-19 could be not only because of vaccination status but also due to the probability of infection in a specific period of time, that depend also from community transmission rates and mitigation measurements. Please discuss this.

These points really be considered. We discussed those together with weakness of the study. 

19. The manuscript needs a grammar English review.

We have asked a native English speaker to review our manuscript and all corrections are in the new version. 

We have also corrected some authors name which were with some type error. The references were renumbered to accommodate 3 new references included in the new version. We have also remove one of the authors which were duplicated in the authorship section. 

Thank you for your attention and I hope our modifications satisfy the reviewers.

Sincerely Yours 

Amilcar Tanuri

UFRJ

---

## [Editor Report · Decision Letter 1]

1 May 2022

PONE-D-22-03333R1COVID-19 cross-sectional study in Maricá, Brazil: the impact of vaccination coverage in viral incidencePLOS ONE

Dear Dr. Tanuri,

Thank you for submitting your manuscript to PLOS ONE. After careful consideration, we feel that it has merit but does not fully meet PLOS ONE’s publication criteria as it currently stands, especially regarding the use of standard English.  Therefore, we invite you to submit a revised version of the manuscript after it has been checked for spelling, grammatical and punctuation errors. For exemple, in the abstract, the following sentences contain erros: "We **made** three cycles of study sampling a total of 59 1134 subjects during May to August 2021"; "...individuals carrying detectable IgG antibodies and neutralizing antibody (NAb) 61 levels was **bigger** in the vaccinated when compared to unvaccinated groups...". We recommend that the manuscript be revised by a native English speaker.

We look forward to receiving your revised manuscript.

Kind regards,

Odir Antonio Dellagostin

Academic Editor

PLOS ONE
---

## [Author Response · Author response to Decision Letter 1]

4 May 2022

Dear Editor 

Thank you for the excellent review done in our manuscript entitled “COVID-19 population-based survey in Maricá, Brazil: the impact of vaccination in viral incidence.” I am sure that reviewers´ comments will make the article clearer and more interesting for readers. 

As advised by the Editor we have asked a native English speaker to review our manuscript and all corrections are in the new version. 

Thank you for your attention and I hope our modifications satisfy the reviewers.

Sincerely Yours 

Amilcar Tanuri

UFRJ

---

## [Editor Report · Decision Letter 2]

13 May 2022

COVID-19 cross-sectional study in Maricá, Brazil: the impact of vaccination coverage in viral incidence

PONE-D-22-03333R2

Dear Dr. Tanuri,

We’re pleased to inform you that your manuscript has been judged scientifically suitable for publication and will be formally accepted for publication once it meets all outstanding technical requirements.

Kind regards,

Odir Antonio Dellagostin

Academic Editor

PLOS ONE
---

## [Editor Report · Acceptance letter]

11 Aug 2022

PONE-D-22-03333R2 

COVID-19 cross-sectional study in Maricá, Brazil: the impact of vaccination coverage on viral incidence 

Dear Dr. Tanuri:

I'm pleased to inform you that your manuscript has been deemed suitable for publication in PLOS ONE. Congratulations! Your manuscript is now with our production department. 

Kind regards, 

on behalf of

Dr. Odir Antonio Dellagostin 

Academic Editor

PLOS ONE